# β-Arrestin-dependent and -independent endosomal G protein activation by the vasopressin type 2 receptor

**Carole Daly[1], Akim Abdul Guseinov[2], Hyunggu Hahn[3,4], Adam Wright[1], Irina G Tikhonova[2], Alex Rojas Bie Thomsen[3,4]\*, Bianca Plouffe[1]\***

[1]Wellcome-Wolfson Institute for Experimental Medicine, School of Medicine, Dentistry and Biomedical Sciences, Queen's University Belfast, Belfast, United Kingdom; [2]School of Pharmacy, Queen's University Belfast, Belfast, United Kingdom; [3]Department of Molecular Pathobiology, New York University College of Dentistry, New York, United States; [4]NYU Pain Research Center, New York University College of Dentistry, New York, United States

**\*For correspondence:**
art8@nyu.edu (ARBieT);
b.plouffe@qub.ac.uk (BP)

**Competing interest:** The authors declare that no competing interests exist.

**Abstract** The vasopressin type 2 receptor ($V_2R$) is an essential G protein-coupled receptor (GPCR) in renal regulation of water homeostasis. Upon stimulation, the $V_2R$ activates $G\alpha_s$ and $G\alpha_{q/11}$, which is followed by robust recruitment of β-arrestins and receptor internalization into endosomes. Unlike canonical GPCR signaling, the β-arrestin association with the $V_2R$ does not terminate $G\alpha_s$ activation, and thus, $G\alpha_s$-mediated signaling is sustained while the receptor is internalized. Here, we demonstrate that this $V_2R$ ability to co-interact with G protein/β-arrestin and promote endosomal G protein signaling is not restricted to $G\alpha_s$, but also involves $G\alpha_{q/11}$. Furthermore, our data imply that β-arrestins potentiate $G\alpha_s/G\alpha_{q/11}$ activation at endosomes rather than terminating their signaling. Surprisingly, we found that the $V_2R$ internalizes and promote endosomal G protein activation independent of β-arrestins to a minor degree. These new observations challenge the current model of endosomal GPCR signaling and suggest that this event can occur in both β-arrestin-dependent and -independent manners.

## eLife assessment

This is an **important** study that contributes to our understanding of the role of beta-arrestins in endosomal activation of the vasopressin type 2 receptors. While the use of a minigene as a tool is a weakness, the evidence is overall **convincing** and makes for significant findings whose theoretical and practical implications extend to other GPCRs.

## Introduction

The vasopressin type 2 receptor ($V_2R$) is mainly known for its antidiuretic action in the kidney. Here, in the principal cells of the collecting duct, the $V_2R$ regulates water reabsorption from pre-urine by promoting translocation of water channel aquaporin 2 (AQP2) located in intracellular vesicles to the apical membrane (*Nielsen et al., 1993*). The net result of this translocation is an enhanced water permeability. Defective $V_2R$ signaling due to loss or gain of function mutations is associated with nephrogenic diabetes insipidus (*Bichet and Bockenhauer, 2016*) or nephrogenic syndrome of inappropriate antidiuresis (*Carpentier et al., 2012*), respectively.

The $V_2R$ belongs to the superfamily of G protein-coupled receptors (GPCRs), membrane proteins that control almost all physiological processes. Canonically, stimulation of GPCRs leads to coupling to

and activation of heterotrimeric G proteins (Gαβγ), which initiates downstream signaling cascades that control global cellular responses. GPCRs can couple to four main families of Gα protein isoforms: Gα$_{s/olf}$, Gα$_{i/o}$, Gα$_{q/11}$, and Gα$_{12/13}$. Activation of each family leads to distinct downstream signaling events and cell biological outcomes. G protein activation is usually short lived and followed by receptor phosphorylation by GPCR kinases, which drives the recruitment of β-arrestins (βarrs) to the phosphorylated receptor. As βarrs interact with the same region of the receptor as G proteins, their recruitment physically uncouples G proteins from the receptor which causes desensitization of G protein signaling (*Pippig et al., 1993*). In addition, βarrs scaffold several proteins involved in endocytosis, which promotes receptor internalization into endosomes (*Krupnick et al., 1997*; *Laporte et al., 1999*).

Surprisingly, recent findings facilitated by the emergence of new molecular tools to interrogate signaling events with a subcellular resolution have challenged this plasma membrane centric view of G protein signaling. Several GPCRs, including the V$_2$R, have been reported to engage in G protein signaling after βarr-mediated receptor internalization into early endosomes and/or other intracellular compartments (*Feinstein et al., 2013*; *Thomsen et al., 2016*). This endosomal stimulation of G protein signaling by βarr-bound GPCRs has been difficult to reconcile with the aforementioned canonical understanding of GPCR signaling since G protein and βarr interactions with a single receptor were thought to be mutually exclusive. However, we discovered and delineated a new signaling paradigm whereby some GPCRs, including the V$_2$R, bind βarrs in a specific manner; in this conformation, also called the 'tail' conformation, βarr only interacts with the phosphorylated receptor carboxy-terminal tail while the transmembrane core of the receptor remains unoccupied (*Cahill et al., 2017*; *Shukla et al., 2014*). Interestingly, βarr in this tail conformation can promote certain βarr-mediated functions such as receptor internalization and signaling (*Cahill et al., 2017*; *Kumari et al., 2016*; *Kumari et al., 2017*). However, as βarr does not compete for the receptor G protein-binding site in this tail conformation, it does not desensitize G protein signaling (*Cahill et al., 2017*). Rather, the receptor in this conformation may interact simultaneously with G proteins and βarr to form a GPCR–G protein–βarr 'megaplex' (*Cahill et al., 2017*; *Nguyen et al., 2019*; *Thomsen et al., 2016*). Thus, the simultaneous engagement with G protein and βarr allows the receptor in these megaplexes to maintain its ability to activate G protein signaling, even while being internalized into endosomes by βarrs. Noteworthy, endosomal Gα$_s$ signaling by V$_2$R has been associated with enhanced and sustained translocation of AQP2 to the plasma membrane to facilitate water reabsorption, and therefore, appears to play an important physiological role (*Feinstein et al., 2013*).

Although known as a Gα$_s$-coupled receptor, several studies report activation of the Gα$_{q/11}$ isoforms by V$_2$R (*Avet et al., 2022*; *Heydenreich et al., 2022*; *Lykke et al., 2015*; *Zhu et al., 1994*) as well as unproductive coupling to Gα$_{12}$ (*Okashah et al., 2020*). Therefore, we hypothesized that the V$_2$R form megaplexes with both Gα$_s$ and Gα$_{q/11}$ leading to endosomal activation of both Gα$_s$ and Gα$_{q/11}$. In addition, pulse-stimulation experiments of the V$_2$R and parathyroid hormone type 1 receptor (PTHR) demonstrated that sustained Gα$_s$-mediated signaling was enhanced by βarr (*Feinstein et al., 2011*; *Feinstein et al., 2013*). To address whether such βarr-mediated increase in G protein signaling is a result of direct coupling and activation of G proteins at endosomes, we here applied a combination of approaches based on engineered mini G proteins (mG proteins) (*Nehmé et al., 2017*; *Wan et al., 2018*), enhanced bystander bioluminescence resonance energy transfer (EbBRET) (*Namkung et al., 2016*), nanoluciferase binary technology (NanoBiT) (*Dixon et al., 2016*), and confocal microscopy imaging.

## Results

### The V$_2$R activates Gα$_s$ and Gα$_q$ from early endosomes

To measure the activation of the four families of G protein isoforms at the plasma membrane and early endosomes by the V$_2$R in real time, we used mG proteins. The mG proteins are homogenously distributed in the cytosol under basal condition but translocate to the subcellular location of active GPCRs upon stimulation (*Carpenter and Tate, 2016*; *Nehmé et al., 2017*; *Wan et al., 2018*). In addition, we applied an EbBRET approach instead of a conventional bioluminescence resonance energy transfer (BRET)-based assay to monitor mG protein trafficking. EbBRET displays superior robustness and sensitivity, as well as higher dynamic spectrometric energy transfer signals compared to conventional BRET, which is why this approach was favored (*Namkung et al., 2016*). We fused mG proteins

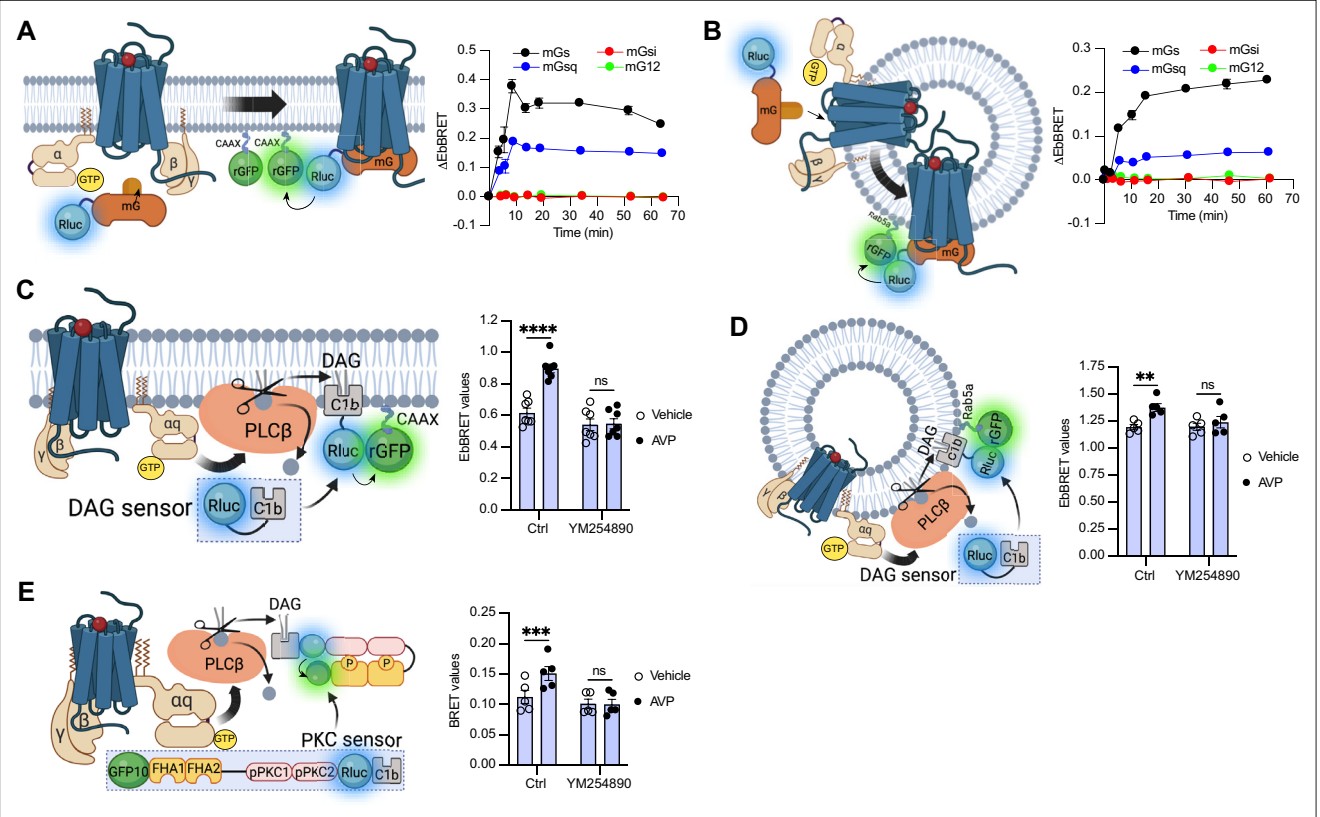

**Figure 1.** Activation of Gα$_s$ and Gα$_q$ at plasma membrane and early endosomes by vasopressin type 2 receptor (V$_2$R) monitored by bioluminescence resonance energy transfer (BRET). (**A**) Left: Illustration of enhanced bystander bioluminescence resonance energy transfer (EbBRET) biosensors used to monitor G protein activation at the plasma membrane. Right: Kinetics of the recruitment of mG proteins at the plasma membrane upon stimulation of V$_2$R with 1 µM arginine vasopressin (AVP) or vehicle. (**B**) Left: Illustration of EbBRET biosensors used to monitor G protein activation at the early endosomes. Right: Kinetics of the recruitment of mG proteins to early endosomes upon stimulation of V$_2$R with 1 µM AVP. (**C**) Left: Illustration of the BRET-based biosensor used to monitor diacylglycerol (DAG) production at the plasma membrane. Right: DAG generated at the plasma membrane upon a 10-min stimulation of V$_2$R with 1 µM AVP or vehicle in cells pre-treated 30 min with 0.1 µM YM254890 or vehicle. (**D**) Left: Illustration of the BRET-based biosensor used to monitor DAG production at the early endosomes. Right: DAG generated at early endosomes upon a 10-min stimulation of V$_2$R with 1 µM AVP in cells pre-treated 30 min with 0.1 µM YM254890 or vehicle. (**E**) Left: Illustration of the BRET-based protein kinase C (PKC) biosensor used to monitor PKC activation. Right: Activation of PKC upon 10 min of stimulation of V$_2$R with 1 µM AVP. For the kinetics, $n$ = 3 (mGs, mGsi, and mGsq) or $n$ = 4 (mG12) independent experiments. For the experiments using the DAG biosensor, $n$ = 7 and $n$ = 5 for measurements at plasma membrane and early endosomes, respectively. For the experiments using the PKC biosensor, $n$ = 5. Asterisks mark statistically significant differences between vehicle and AVP treatments as assessed by two-way analysis of variance (ANOVA) and Sidak's post hoc test for multiple comparisons (**$p \le 0.01$, ***$p \le 0.001$, ****$p \le 0.0001$). Data are shown as mean ± standard error on mean.

The online version of this article includes the following source data and figure supplement(s) for figure 1:

**Source data 1.** Raw data on *Figure 1*.

**Figure supplement 1.** Equivalent expression of mG constructs for the vasopressin type 2 receptor (V$_2$R) kinetics.

to the luciferase from *Renilla reniformis* (Rluc) and anchored green fluorescent protein from the same species (rGFP) to the polybasic sequence and prenylation CAAX box of KRas (rGFP-CAAX), which is located at the plasma membrane (*Zacharias et al., 2002*), or to the early endosome marker Rab5 (*Gorvel et al., 1991*; *Figure 1A, B*, left panels). Four variants of mG proteins (mGs, mGsi, mGsq, and mG12) have been designed and shown to maintain the receptor-Gα protein specificity of the four Gα subunit isoform families (*Wan et al., 2018*). In HEK293 cells expressing rGFP-CAAX, V$_2$R, and similar levels of Rluc-fused mG proteins (*Figure 1—figure supplement 1A*), arginine vasopressin (AVP) treatment induced a rapid recruitment of mGs and mGsq but not mGsi nor mG12 to the plasma membrane. Maximal recruitment of the mGs and mGsq were reached ~10 min after initial stimulation (*Figure 1A*, right panel). These results suggest that the V$_2$R activates both Gα$_s$ and Gα$_{q/11}$ at the plasma membrane. In addition, AVP stimulation led to the recruitment of the same mG protein isoforms to

early endosomes in cells expressing rGFP-Rab5, $V_2R$, and similar levels of Rluc-fused mG proteins (*Figure 1B*, right panel, *Figure 1—figure supplement 1B*). In contrast to the plasma membrane response, mGs and mGsq recruitment to early endosomes were slower and reached maximal levels 45–60 min after initial stimulation with AVP (*Figure 1B*, right panel).

Activation of $G\alpha_{q/11}$ canonically stimulates phospholipase C$\beta$ (PLC$\beta$), an enzyme that hydrolyzes membrane phosphoinositides into diacylglycerol (DAG) and inositol phosphate. Consequently, to validate $G\alpha_{q/11}$ activation at plasma membrane and early endosomes by the $V_2R$, we monitored the AVP-mediated production of DAG at these subcellular compartments. To monitor DAG at the plasma membrane, we used the C1b DAG-binding domain of PKC$\delta$ fused to Rluc (Rluc-C1b) and measured the EbBRET between Rluc-C1b and rGFP-CAAX (*Wright et al., 2021*) upon stimulation with AVP or vehicle (*Figure 1C*, left panel). In line with the recruitment of mGsq at the plasma membrane observed in *Figure 1A*, AVP induced a recruitment of Rluc-C1b at the plasma membrane depicted by an increase of EbBRET values, which confirms the production of DAG at the plasma membrane by $V_2R$ (*Figure 1C*, right panel). Importantly, this production of DAG is $G\alpha_{q/11}$-dependent as a pre-treatment with YM254890, an inhibitor of $G\alpha_{q/11}$, abrogated this downstream response (*Taniguchi et al., 2003*). To measure the production of DAG at early endosomes by $V_2R$, we monitored the EbBRET between Rluc-C1b and rGFP-Rab5 upon $V_2R$ stimulation (*Figure 1D*, left panel). Similar to the observed recruitment of mGsq to the early endosomes (*Figure 1B*), AVP also induced a recruitment of Rluc-C1b to the early endosomal marker rGFP-Rab5, which suggests that DAG is produced at endosomal membranes (*Figure 1D*, right panel). This endosomal DAG production was also inhibited by YM254890 treatment, which demonstrates that it is $G\alpha_{q/11}$ dependent (*Figure 1D*, right panel). As DAG is an activator of protein kinase C (PKC), we further monitored PKC activation upon stimulation with AVP or vehicle. To measure PKC activation, we used an unimolecular BRET-based sensor composed of two Rluc-C1b-fused PKC consensus sequences (TLKI;pPKC1 and TLKD;pPKC2) that are connected to the phosphothreonine-binding domains FHA1 and FHA2 and the BRET acceptor GFP10 via a flexible linker (*Namkung et al., 2018*; *Figure 1E*, left panel). Upon phosphorylation of pPKC1 and pPKC2, FHA1 and FHA2 bind to these phosphorylated sequences resulting in an increased proximity between Rluc and GFP10 and a corresponding increase of BRET signal. In line with the AVP-mediated production of DAG, AVP treatment also induced phosphorylation of PKC in a $G\alpha_{q/11}$-dependent fashion (*Figure 1E*, right panel).

To visualize $V_2R$-mediated activation of $G\alpha_s$ and $G\alpha_{q/11}$ at the plasma membrane and early endosomes we used confocal microscopy. For this purpose, we transfected HEK293 cells with mGs, mGsq, or mGsi (as negative control) fused to a HaloTag (Halo-mGs, Halo-mGsq, Halo-mGsi) along with the respective red fluorescent protein (RFP)-fused plasma membrane or early endosomes markers Lck (*Ley et al., 1994*) or early endosome antigen 1 (EEA1) (*Simonsen et al., 1998*). Upon HaloTag labeling with a fluorescent green ligand, mGs, mGsq, and mGsi were visible and homogenously distributed in the cytosol under basal condition (vehicle) (*Figure 2A*, upper panels). In contrast, in cells treated with AVP for 10 min, mGs and mGsq but not mGsi were redistributed along the periphery of the cells where they colocalized with Lck (*Figure 2A*, bottom panels, *Figure 2B*). These observations confirm that $G\alpha_s$ and $G\alpha_{q/11}$ are activated by the $V_2R$ at the plasma membrane. In cells expressing the early endosomal marker EEA1, robust colocalization between Halo-mGs, Halo-mGsq and RFP-EEA1 were found 45 min after initial AVP stimulation but not by vehicle treatment (*Figure 2C, D*). Together our EbBRET and confocal microscopy imaging data suggest that $G\alpha_s$ and $G\alpha_{q/11}$ are activated by $V_2R$ first at plasma membrane, and later on, from early endosomes after the $V_2R$ has been internalized.

## The $V_2R$ recruits $G\alpha_s$/$G\alpha_q$ and $\beta$arrs simultaneously

G protein activation from endosomes by some GPCRs is associated with the ability of the receptor to recruit G protein and $\beta$arr simultaneously to form a GPCR–$\beta$arr–G protein megaplex. As we already previously demonstrated that the $V_2R$ forms $V_2R$-$\beta$arr-$G_s$ megaplexes upon AVP stimulation (*Thomsen et al., 2016*), we here explored whether formation of such complexes potentially can be formed with $G_{q/11}$ as well. In addition to the $V_2R$, we also applied a chimeric $V_2R$ harboring the carboxy-terminal tail of the $\beta_2$-adrenergic receptor ($\beta_2AR$) referred to as $V_2\beta_2AR$. We previously showed that the phosphorylated $V_2R$ carboxy-tail forms stable complexes with $\beta$arr, a requirement of megaplex formation, whereas the carboxy-tail of the $\beta_2AR$ does not (*Cahill et al., 2017*). Therefore, we expected that only the $V_2R$, but not the $V_2\beta_2AR$, recruits G proteins and $\beta$arrs simultaneously upon agonist challenge.

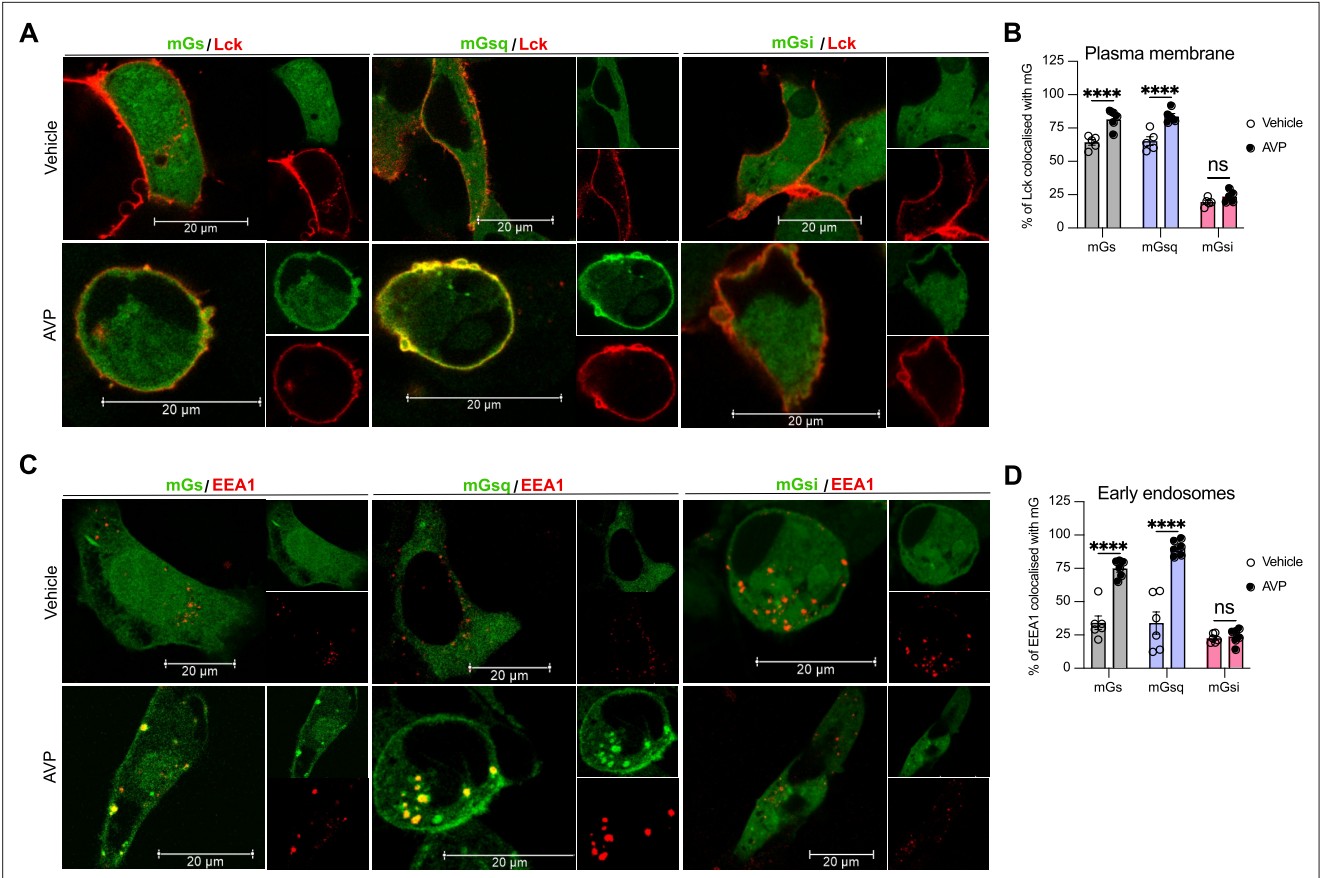

**Figure 2.** Activation of Gα_s and Gα_q at plasma membrane and early endosomes by vasopressin type 2 receptor (V_2R) monitored by confocal microscopy. (**A**) Representative confocal microscopy images of cells expressing red fluorescent protein (RFP)-Lck, V_2R, and Halo-mGs (left panels), Halo-mGsq (middle panels), or Halo-mGsi (right panels) stimulated for 10 min with vehicle (upper panels) or 1 μM arginine vasopressin (AVP) (bottom panels). (**B**) Percentages of Lck colocalized with each Halo-mG calculated from five representative images. (**C**) Representative confocal microscopy images of cells expressing RFP-early endosome antigen 1 (EEA1), V_2R, and Halo-mGs (left panels), Halo-mGsq (middle panels), or Halo-mGsi (right panels) stimulated for 45 min with vehicle (upper panels) or 1 μM AVP (bottom panels). (**D**) Percentages of EEA1 colocalized with each Halo-mG calculated from six representative images. $n = 3$ independent experiments for Lck and EEA1. Statistical differences between vehicle and AVP treatments were assessed by two-way analysis of variance (ANOVA) and Sidak's post hoc test for multiple comparisons (****$p \leq 0.0001$). Data are shown as mean ± standard error on mean.

The online version of this article includes the following source data for figure 2:

**Source data 1.** Raw data on *Figure 2*.

Both the V_2R and V_2β_2AR bind to AVP with similar affinities and activate adenylyl cyclase via Gα_s with similar potencies (*Oakley et al., 1999*). We monitored activation of the four Gα protein families at the plasma membrane by the V_2β_2AR upon AVP treatment using the same approach utilized in *Figure 1A*. Similar to the V_2R, the V_2β_2AR activated both Gα_s and Gα_q, but not Gα_i nor Gα_12, at plasma membrane with a maximal response reached after ~10 min of stimulation (*Figure 3A*, *Figure 3—figure supplement 1*). While the V_2R and V_2β_2AR are both reported to internalize via a βarr-dependent mechanism, βarr has been reported to rapidly dissociate from the V_2β_2AR shortly after its recruitment to the plasma membrane due to its low affinity for this receptor chimera (*Oakley et al., 1999*). In contrast, βarr stays associated with the V_2R during its internalization into endosomes owning to its high affinity for the V_2R (*Oakley et al., 1999*). Here, we compared the kinetics of βarr1 and βarr2 recruitment to the V_2R and V_2β_2AR at the plasma membrane and early endosomes by monitoring AVP-promoted EbBRET between Rluc-fused βarrs and rGFP-CAAX or rGFP-Rab5, respectively (*Figure 3B*, left panel). At similar levels of receptor and βarr expressions (*Figure 3—figure supplement 2*), both receptors recruited βarr1 and βarr2 at plasma membrane maximally after 10 min of stimulation with AVP (*Figure 3B*, right upper panel). However, the presence of βarrs at the plasma membrane declined rapidly hereafter 10 min in

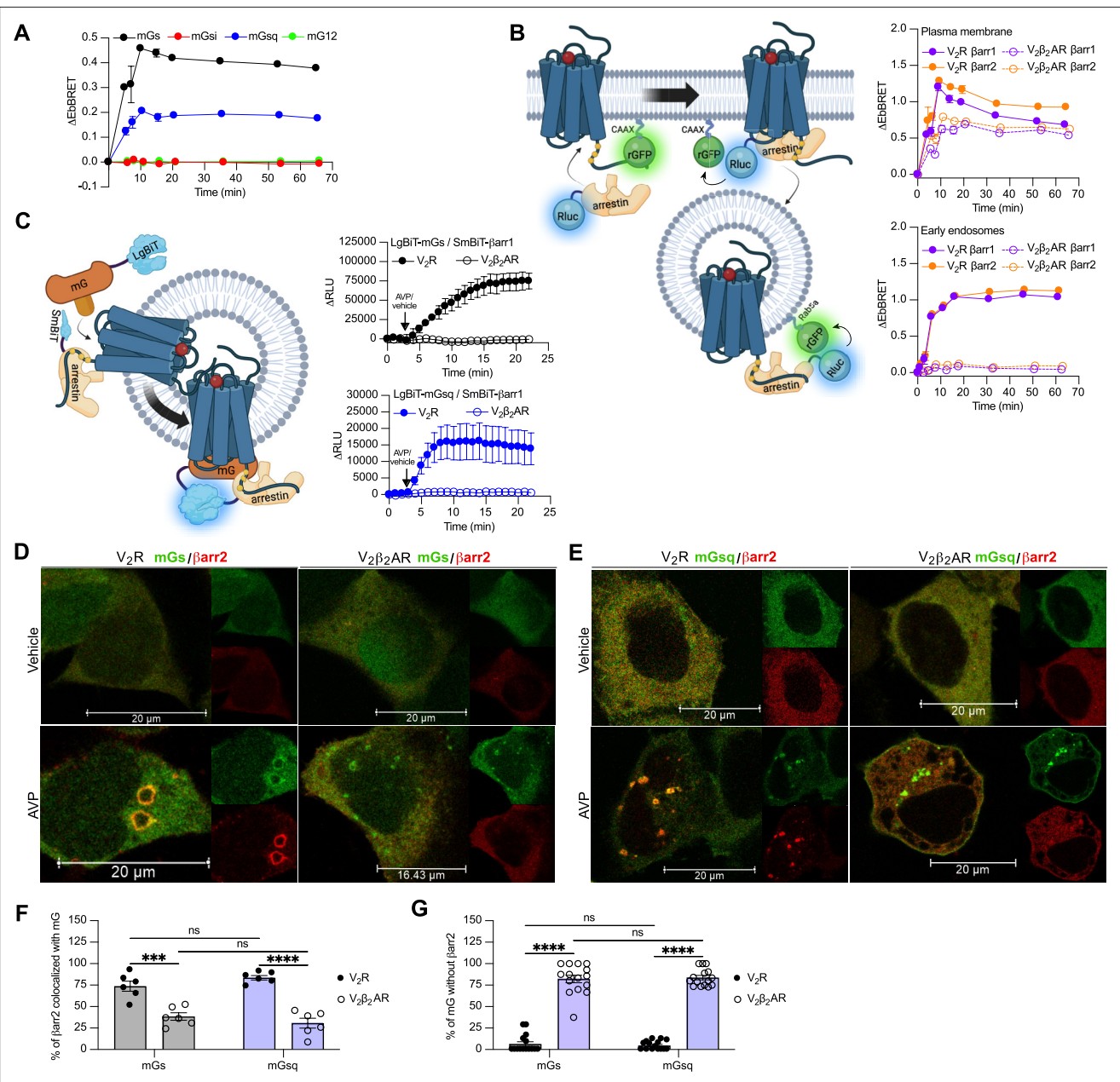

**Figure 3.** Formation of megaplexes with Gα$_s$ or Gα$_q$ upon stimulation with arginine vasopressin (AVP). (**A**) Kinetics of the recruitment of mG proteins at the plasma membrane upon stimulation of V$_2$β$_2$AR with 1 µM AVP. $n = 3$ for mGs, mGsi, and mGsq, and $n = 4$ for mG12. (**B**) Left: Illustration of the enhanced bystander bioluminescence resonance energy transfer (EbBRET) biosensors used to monitor βarr recruitment to the plasma membrane and early endosomes. Right: Kinetics of the recruitment of βarr1 and βarr2 to the plasma membrane (upper panel) and to early endosomes (bottom panel) upon stimulation of vasopressin type 2 receptor (V$_2$R) or V$_2$β$_2$AR with 1 µM AVP. $n = 3$ for all conditions. (**C**) Left panel: Illustration of the nanoBiT biosensors used to monitor simultaneous coupling of Gα proteins and βarr1 to G protein-coupled receptors (GPCRs). Right panels: Kinetics of the proximity between SmBiT-βarr1 and LgBiT-mGs (upper panel) or LgBiT-mGsq (bottom panel) upon stimulation of the receptors with 1 µM AVP. $n = 3$ for all conditions. (**D**) Representative confocal microscopy images of cells expressing Halo-mGs, strawberry-βarr2, and V$_2$R (left panels), or V$_2$β$_2$AR (right panels) and stimulated for 45 min with vehicle (upper panels) or 1 µM AVP (bottom panels). (**E**) Representative confocal microscopy images of cells expressing Halo-mGsq, strawberry-βarr2, and V$_2$R (left panels), or V$_2$β$_2$AR (right panels) and stimulated for 45 min with vehicle (upper panels) or 1 µM AVP (bottom panels). (**F**) Percentage of βarr2 colocalization with mGs or mGsq upon stimulation with 1 µM AVP (six representative images). (**G**) Percentage of mGs or mGsq puncta observed that were not colocalized with βarr2 upon stimulation with 1 µM AVP (15 representative images). Asterisks mark significant differences between V$_2$R and V$_2$β$_2$AR assessed by two-way analysis of variance (ANOVA) and Sidak's post hoc test for multiple comparisons (***$p \leq 0.001$, ****$p \leq 0.0001$). No statistical difference (ns) was detected between mGs and mGsq. Data are shown as mean ± standard error on mean.

The online version of this article includes the following source data and figure supplement(s) for figure 3:

*Figure 3 continued on next page*

*Figure 3 continued*

**Source data 1.** Raw data on *Figure 3*.

**Figure supplement 1.** Equivalent expression of mG constructs for the $V_2\beta_2AR$ kinetics.

**Figure supplement 2.** Relative expression of vasopressin type 2 receptor ($V_2R$) and $V_2\beta_2AR$ at the plasma membrane and of βarrs.

$V_2R$-expressing cells, while remaining for longer periods of time in $V_2\beta_2AR$-expressing cells. These findings are in line with the previous reported observations (*Oakley et al., 1999*). Additionally, the translocation of βarrs to the plasma membrane was more robust for the $V_2R$ as compared to the $V_2\beta_2AR$, which is reminiscent from the higher affinity of βarrs for the $V_2R$ as compared to the $\beta_2AR$ (*Oakley et al., 2000*). In contrast to the rapid translocation of βarrs to the plasma membrane, AVP treatment induced a robust but slower enrichment of βarrs to early endosomes with $V_2R$ reaching a maximal response after approximately 45 min of stimulation (*Figure 3B*, right bottom panel). Importantly, as opposed to $V_2R$, AVP stimulation of $V_2\beta_2AR$ did not result in βarr translocation to early endosomes (*Figure 3B*, right bottom panel). Consequently, the $V_2\beta_2AR$ represents a valuable negative control to investigate the ability to recruit G proteins and βarrs simultaneously at endosomes.

To track the simultaneous coupling of G proteins and βarrs to GPCRs in real time, a nanoBiT approach was used. Both mGs and mGsq were fused to the large portion of nanoluciferase (large-BiT; LgBiT) and βarr1 to an optimized small peptide BiT (small BiT; SmBiT). Reconstitution of the complete and functional nanoluciferase, which catalyzes the conversion of coelenterazine-h, results in emission of a bright luminescence signal. In our setup, close proximity of LgBiT and SmBiT only occurs when LgBiT-mG and SmBiT-βarr1 are recruited simultaneously to the receptor, which is a hallmark of megaplex formation (*Figure 3C*, left panel). Using this approach, we detected bright luminescence signals involving mGs/βarr1 (*Figure 3C*, upper right panel) and mGsq/βarr1 (*Figure 3C*, bottom right panel) upon stimulation of the $V_2R$ but not the $V_2\beta_2AR$. Interestingly, the dual coupling of $G\alpha_{q/11}$/βarr to $V_2R$ appeared to be faster than the co-coupling of $G\alpha_s$/βarr. While 20 min was required to reach the maximal response of $V_2R$-stimulated mGs/βarr1 co-coupling, 8 min was sufficient to obtain the maximal levels of mGsq/βarr1 recruitment to the $V_2R$ (*Figure 3C*, right panels).

To visualize the simultaneous recruitment of G proteins and βarr to the receptors by confocal microscopy, we transfected HEK293 cells with βarr2 fused to strawberry (strawberry-βarr2), Halo-mGs or Halo-mGsq, and the $V_2R$ or $V_2\beta_2AR$. In vehicle-treated cells, both mGs and βarr2 were homogenously distributed in the cytosol (*Figure 3D*, upper panels). However, after prolonged stimulation of $V_2R$ with AVP, around 75% of βarr2 colocalized with mGs in endocytic vesicles (*Figure 3D, F*). In $V_2\beta_2AR$-stimulated cells, little to no colocalization was observed between βarr2 and mGs upon prolonged stimulation with AVP (*Figure 3D, F*). Surprisingly, however, some clusters of intracellular mGs were visible despite the absence of βarr2 (*Figure 3D*). As opposed to $V_2R$-expressing cells for which nearly all the clusters of intracellular mGs colocalize with βarr2, in $V_2\beta_2AR$-expressing cells around 75% of the intracellular mGs clusters do not colocalize with βarr2 (*Figure 3G*). These results suggest simultaneous coupling of $G\alpha_s$/βarr2 to the $V_2R$ in endosomes but not to the $V_2\beta_2AR$. It is important to specify here that as mG proteins are recruited to receptors in the active conformation, the mG clusters are located where the active receptors are. Consequently, colocalization of mGs with βarr2 necessarily implies the presence of the active receptor in close proximity to both mGs and βarr2. In cells expressing mGsq, both mGsq and βarr2 were also homogenously distributed in the cytosol when cells were treated with the vehicle (*Figure 3E*, upper panels). Upon AVP stimulation, approximately 75% of βarr2 colocalized with mGsq in intracellular vesicles in $V_2R$-expressing cells, whereas poor colocalization was observed in $V_2\beta_2AR$-expressing cells (*Figure 3E, F*). Similar to cells expressing mGs, clear clusters of intracellular mGsq were visible in cells expressing $V_2\beta_2AR$ (*Figure 3E, G*), suggesting a certain level of endosomal $G\alpha_s$/$G\alpha_q$ signaling despite the absence of βarr2.

## βarr-dependent and -independent endosomal G protein activation by the $V_2R$

Activation of $G\alpha_s$ and $G\alpha_{q/11}$ by the $V_2\beta_2AR$ from endosome-like structures in the absence of local βarrs raises the possibility that the $V_2R$ can activate these G proteins from endosomes in both βarr-dependent and -independent manners. To test this hypothesis, we compared AVP-induced $G\alpha_s$ and $G\alpha_{q/11}$ activation at plasma membrane and endosomes in CRISPR/Cas9-engineered βarr1- and

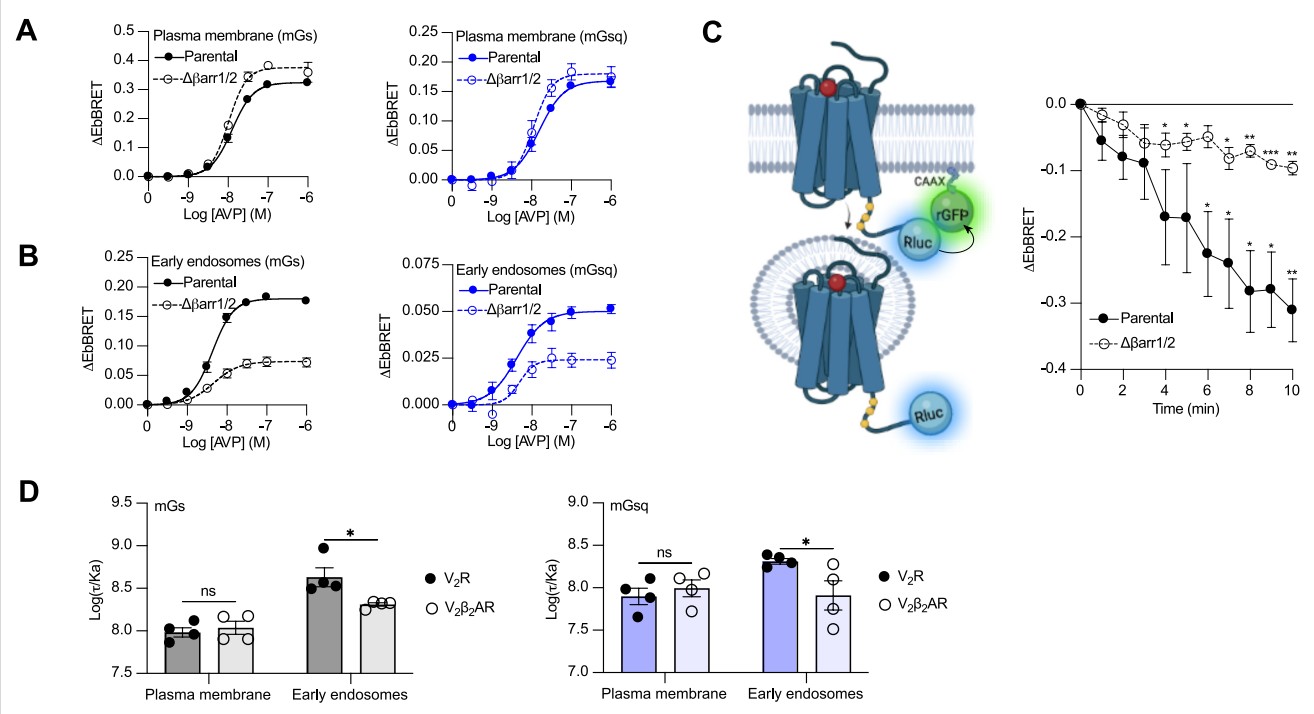

**Figure 4.** Contribution of megaplex to endosomal Gα$_s$ and Gα$_q$ signaling. (**A**) Arginine vasopressin (AVP) dose–response curves of the recruitment of mGs (left panel) and mGsq (right panel) to the plasma membrane in parental and Δβarr1/2 cells expressing vasopressin type 2 receptor (V$_2$R) upon 10 min of stimulation. (**B**) AVP dose–response curves of the recruitment of mGs (left panel) and mGsq (right panel) to early endosomes in parental and Δβarr1/2 cells expressing V$_2$R upon 45 min of stimulation. (**C**) Left: Illustration of enhanced bystander bioluminescence resonance energy transfer (EbBRET) biosensors used to monitor AVP-mediated internalization of the V$_2$R. Right: Kinetics of V$_2$R internalization upon stimulation with AVP 0.1 µM. Asterisks mark significant differences from zero as assessed by one sample t test (*p ≤ 0.05, **p ≤ 0.01, ***p ≤ 0.001). (**D**) Transduction coefficients of mGs (left panel) or mGsq (right panel) recruitment to the plasma membrane and early endosomes in V$_2$R- or V$_2$β$_2$AR-expressing cells. Asterisks mark significant differences between the V$_2$R and V$_2$β$_2$AR as assessed by two-way analysis of variance (ANOVA) and Sidak's post hoc test for multiple comparisons (*p ≤ 0.05). No statistical difference (ns) was detected between the V$_2$R and V$_2$β$_2$AR at the plasma membrane. n = 4 biological replicates for all experiments. Data are shown as mean ± standard error on mean.

The online version of this article includes the following source data and figure supplement(s) for figure 4:

**Source data 1.** Raw data on **Figure 4**.

**Figure supplement 1.** Relative expression of vasopressin type 2 receptor (V$_2$R) at the plasma membrane in parental and Δβarr1/2 cells.

**Figure supplement 2.** Relative expression of vasopressin type 2 receptor (V$_2$R)-Rluc in parental and Δβarr1/2 cells.

**Figure supplement 3.** Relative expression of vasopressin type 2 receptor (V$_2$R) and V$_2$β$_2$AR at the plasma membrane.

**Figure supplement 4.** Arginine vasopressin (AVP) dose–response curves of the recruitment of mGs and mGsq to the plasma membrane and early endosomes by the vasopressin type 2 receptor (V$_2$R) and V$_2$β$_2$AR.

βarr2-deficient HEK293 cells (Δβarr1/2) (**Namkung et al., 2016**) as well as their parental cellular counterpart. The surface expression of V$_2$R was matched in both cellular backgrounds (**Figure 4—figure supplement 1**). Using the EbBRET biosensors described in **Figure 1A, B**, we performed AVP concentration–response characterization of Gα$_s$ and Gα$_{q/11}$ activation at the plasma membrane (**Figure 4A**) and early endosomes (**Figure 4B**) in parental and Δβarr1/2 HEK293 cells. In contrast to Gα$_s$ and Gα$_{q/11}$ activation at the plasma membrane, which were not negatively affected by the absence of βarrs (**Figure 4A**, **Table 1**), we observed a robust decrease in the ability of the V$_2$R to activate Gα$_s$ and Gα$_q$ at endosomes in Δβarr1/2 cells as compared to their parental counterpart (**Figure 4B**, **Table 1**). These data demonstrate the important role of βarrs in endosomal Gα$_s$/Gα$_{q/11}$ activation by the V$_2$R. However, although the Δβarr1/2 cells do not express βarrs, we still observed significant residual G protein activation from endosomes. This surprising observation suggests that the V$_2$R internalizes into endosomes to some extent in a βarr-independent manner from where G proteins are stimulated. To probe this possibility, we compared V$_2$R internalization in parental and Δβarr1/2 HEK293 cells expressing rGFP-CAAX and equivalent amounts of V$_2$R fused to Rluc at its carboxy-terminal tail (V$_2$R-Rluc) (**Figure 3C**,

**Table 1.** Parameters related to AVP dose–response curves of the mGs and mGsq recruitment to the plasma membrane or early endosomes in parental and Δβarr1/2 cells.

| | | Arginine vasopressin (AVP)-induced maximal efficacy (ΔEbBRET ± SEM) | Potency (LogEC$_{50}$ ± SEM) |
|---|---|---|---|
| Plasma membrane | mGs parental | 0.324 ± 0.006 | −7.90 ± 0.03 |
| | mGs Δβarr1/2 | 0.376 ± 0.011** | −7.99 ± 0.04 |
| | mGsq parental | 0.168 ± 0.005 | −7.81 ± 0.04 |
| | mGsq Δβarr1/2 | 0.180 ± 0.009 | −7.95 ± 0.07 |
| Early endosomes | mGs parental | 0.180 ± 0.004 | −8.37 ± 0.03 |
| | mGs Δβarr1/2 | 0.074 ± 0.004**** | −8.34 ± 0.09 |
| | mGsq parental | 0.050 ± 0.002 | −8.38 ± 0.08 |
| | mGsq Δβarr1/2 | 0.024 ± 0.002**** | −8.32 ± 0.12 |

ΔEbBRET values were fitted using four parameters equation with the bottom fixed at zero. $n = 4$ biological replicates for each condition. Statistical differences between parental and Δβarr1/2 cells for AVP-induced maximal efficacy and potency were assessed by comparing independent fits with a global fit that shares the selected parameter using extra sum-of-squares $F$-test (**$p ≤ 0.01$, ****$p ≤ 0.0001$).

left panel, *Figure 4—figure supplement 2*). In parental HEK293 cells, AVP stimulation of the V$_2$R-Rluc led to a robust decrease of EbBRET values, which indicates strong receptor internalization (*Figure 4C*, right panel). Interestingly, we also observed significant internalization of the V$_2$R in Δβarr1/2 HEK293 cells, although less than in the parental cells (*Figure 4C*, right panel). These results suggest that a minor population of V$_2$R internalizes independently of βarrs and contributes to endosomal Gα$_s$ and Gα$_{q/11}$ signaling.

## βarrs potentiate endosomal Gα$_s$ and Gα$_q$ activation by the V$_2$R

Although our data suggest that a minor population of V$_2$R internalizes in the absence of βarrs and contribute to V$_2$R-mediated endosomal Gα$_s$ signaling, it has been reported that βarr binding to the V$_2$R and PTHR potentiates endosomal Gα$_s$ signaling (*Feinstein et al., 2011*; *Feinstein et al., 2013*). To verify this and determine if this potentiator effect of βarrs also affects endosomal Gα$_{q/11}$ signaling, we compared endosomal Gα$_s$ and Gα$_{q/11}$ activation in cells expressing similar levels of V$_2$R or V$_2$β$_2$AR (*Figure 4—figure supplement 3*). Our rationale for using these two receptors is that if βarrs potentiate endosomal G protein activation, this potentiator effect will be observed to a greater extent for the V$_2$R since this receptor associates more robustly with βarrs as compared to the V$_2$β$_2$AR. Using the same biosensors as in *Figure 1A, B*, we performed AVP dose–response curves of mGs and mGsq recruitment to the plasma membrane and early endosomes (*Figure 4—figure supplement 4*, *Table 2*). From

**Table 2.** Parameters related to AVP dose–response curves of the recruitment of mGs and mGsq to the plasma membrane and early endosomes by the V$_2$R or V$_2$β$_2$AR.

| | | Arginine vasopressin (AVP)-mediated maximal efficacy (ΔEbBRET ± SEM) | Potency (logEC$_{50}$ ± SEM) | Transduction coefficient (log(τ/Ka) ± SEM) |
|---|---|---|---|---|
| mGs | PM V$_2$R | 0.300 ± 0.018 | −8.07 ± 0.10 | 7.98 ± 0.05 |
| | PM V$_2$β$_2$AR | 0.392 ± 0.025*** | −8.07 ± 0.10 | 8.04 ± 0.08 |
| | EE V$_2$R | 0.167 ± 0.008 | −8.65 ± 0.09 | 8.63 ± 0.11 |
| | EE V$_2$β$_2$AR | 0.150 ± 0.003 | −8.34 ± 0.04** | 8.31 ± 0.02* |
| mGsq | PM V$_2$R | 0.161 ± 0.011 | −7.88 ± 0.11 | 7.90 ± 0.10 |
| | PM V$_2$β$_2$AR | 0.183 ± 0.013 | −7.97 ± 0.11 | 7.99 ± 0.10 |
| | EE V$_2$R | 0.050 ± 0.002 | −8.44 ± 0.08 | 8.31 ± 0.03 |
| | EE V$_2$β$_2$AR | 0.061 ± 0.005* | −7.82 ± 0.14*** | 7.91 ± 0.17* |

ΔEbBRET values from the dose–response curves were fitted using four parameters equation with the bottom fixed at zero. $n = 4$ biological replicates for each condition. Statistical differences between V$_2$R and V$_2$β$_2$AR for AVP-mediated maximal efficacy and potency were assessed by comparing independent fits with a global fit that shares the selected parameter using extra sum-of-squares $F$-test (*$p ≤ 0.05$, **$p ≤ 0.01$, ***$p ≤ 0.001$). Statistical differences between V$_2$R and V$_2$β$_2$AR for log(τ/Ka) values were assessed by two-way analysis of variance (ANOVA) and Sidak's post hoc test for multiple comparisons (*$p ≤ 0.05$). PM, plasma membrane; EE, early endosomes.

the dose–response curves obtained (*Figure 4—figure supplement 4*, *Table 2*), we determined the transduction coefficient log(τ/Ka), a parameter that combines efficiency and potency to determine the overall G protein transduction, for each condition using the operational model of Kenakin and Christopoulos (*Kenakin et al., 2012*). In cells expressing mGs, the transduction coefficients of Gα$_s$ activation at the plasma membrane were similar for the V$_2$R and V$_2$β$_2$AR, but higher for the V$_2$R than V$_2$β$_2$AR in early endosomes (*Figure 4D*, left panel, *Table 2*). Similarly, in cells expressing mGsq, the transduction coefficients of Gα$_{q/11}$ activation at the plasma membrane were similar for the V$_2$R and V$_2$β$_2$AR, but higher for the V$_2$R than V$_2$β$_2$AR in early endosomes (*Figure 4D*, right panel, *Table 2*). Altogether these results indicate that βarrs potentiate activation of G proteins by the V$_2$R in early endosomes.

## Discussion

In the present work, we addressed the spatial aspect of G protein signaling by the V$_2$R and investigated the potential role of βarrs in modulating these responses in HEK293 cells. Several studies report activation of Gα$_s$ and Gα$_{q/11}$ by the V$_2$R using a wide range of assays (*Avet et al., 2022*; *Heydenreich et al., 2022*; *Inoue et al., 2019*; *Okashah et al., 2020*; *Zhu et al., 1994*). However, these assays lack spatial resolution or are measured by default at the plasma membrane. Here, we demonstrated that both Gα$_s$ and Gα$_{q/11}$ are activated by the V$_2$R at the plasma membrane and early endosomes using a mG protein-based approach as well as biosensors of downstream signaling. The mG protein-based approach measures the receptor's ability to couple to G proteins in real time rather than G protein-mediated signaling outputs. Whether G protein coupling to the V$_2$R follows similar kinetic patterns in primary cells expressing the receptor endogenously was not interrogated in this study. The PTHR, a GPCR that regulates mineral ion homeostasis and bone development, also couples to both Gα$_s$ and Gα$_{q/11}$ (*Schwindinger et al., 1998*). Similar to our observations of the V$_2$R, the reduction of PTHR internalization by βarr1 and βarr2 depletion strongly decreases endosomal Gα$_s$/cAMP signaling. However, in contrast to the V$_2$R, βarr-mediated receptor internalization shuts down Gα$_{q/11}$-mediated responses, and thus, the PTHR does not appear to stimulate Gα$_{q/11}$ from endosomes (*Castro et al., 2002*; *Feinstein et al., 2011*).

The reason for this inability of internalized PTHR to activate Gα$_{q/11}$ from endosomes is not known. However, it is unlikely to be a general feature of these G protein isoforms as multiple laboratories have reported endosomal GPCR signaling events downstream of Gα$_{q/11}$ activation (*Gorvin et al., 2018*; *Jensen et al., 2017*; *Jimenez-Vargas et al., 2018*). These events include measurements of signal-amplified responses such as PKC recruitment or ERK1/2 activation (*Gorvin et al., 2018*; *Jensen et al., 2017*; *Jimenez-Vargas et al., 2018*). Recently, direct activation of Gα$_{q/11}$ from early endosomes was monitored using a mG protein-based approach and effector membrane translocation assay (EMTA). In this study, Wright et al. demonstrated that stimulation of Gα$_{q/11}$ protein isoforms by receptors at the plasma membrane does not necessarily lead to the activation of the exact same isoforms at endosomes (*Wright et al., 2021*). For example, the authors showed that the thromboxane A$_2$ alpha isoform receptor (TPαR) robustly activates all the Gα$_{q/11}$ isoforms (Gα$_q$, Gα$_{11}$, Gα$_{14}$, and Gα$_{15}$) at the plasma membrane, but only activates Gα$_q$ and Gα$_{11}$ isoforms at endosomes. In contrast, the muscarinic acetylcholine M$_3$ receptor (M$_3$R) activates all four Gα$_{q/11}$ isoforms both at plasma membrane and endosomes. While G protein selectivity at plasma membrane is mainly dependent on receptor conformation (*Rose et al., 2014*; *Van Eps et al., 2018*), specific residues present at the GPCR–Gα protein interface (*Flock et al., 2017*) as well as the location and duration of these intermolecular interactions (*Sandhu et al., 2022*), endosomal G protein activation seems to be controlled by additional factors that are not fully understood.

The presence of serine/threonine phosphorylation site clusters at the carboxy-terminal tail of GPCRs delineates two major classes of receptors: class A and class B (*Oakley et al., 2000*). Class A GPCRs such as the β$_2$AR are defined by harboring few single phosphorylation sites, which form interactions with positively charged residues of βarrs (*Shukla et al., 2013*). In addition to the phosphorylated receptor residues, the class A GPCR–βarr association also depends on an interaction between the βarr fingerloop region and the receptor transmembrane core, which sterically block G protein access to the GPCR (*Cahill et al., 2017*; *Shukla et al., 2014*). The class A GPCR–βarr association is transient and the complex dissociates shortly after endocytosis, which results in receptor recycling back to the cell surface. In contrast, class B GPCRs including the V$_2$R are defined by having phosphorylation site clusters in the carboxy-terminal tail that form highly stable associations with βarrs solely

through this region. This strong interaction leads to prolonged receptor internalization into endosomes (*Cahill et al., 2017*). As the stability of this GPCR–βarr complex 'tail' conformation does not depend on the interaction between the βarr fingerloop region and the receptor core, the GPCR can interact simultaneously with G proteins to forms a GPCR–G protein–βarr megaplex (*Nguyen et al., 2019*; *Thomsen et al., 2016*). As the receptors in these megaplexes maintain their ability to activate G proteins, they can internalize via βarrs into different intracellular compartments while stimulating G protein signaling for prolonged periods of time (*Cahill et al., 2017*; *Calebiro et al., 2009*; *Feinstein et al., 2013*; *Ferrandon et al., 2009*; *Thomsen et al., 2016*). Previously, formation of megaplexes at intracellular compartments has only been reported involving $G\alpha_s$ or $G\alpha_{i/o}$ proteins (*Hahn et al., 2022*; *Smith et al., 2021*; *Thomsen et al., 2016*). In the present study, we demonstrate that megaplex formation is not confined to these G protein isoforms but also appears to form with other G protein isoforms such as $G\alpha_{q/11}$.

An interesting aspect of βarr/megaplex-dependent endosomal G protein signaling is whether βarrs only acts as a vehicle that transports GPCRs to this subcellular location from where they activate G proteins or whether βarrs in megaplexes themselves directly modulate G protein activity. In the current study, we show that βarrs directly potentiate G protein activation by the $V_2R$ in early endosomes (*Figure 4D*). These findings are further supported by Feinstein et al. who previously demonstrated that $V_2R$-stimulated G protein activation is positively modulated by the presence of βarr2 (*Feinstein et al., 2013*). However, in the recent cryo-electron microscopy high-resolution structure of an engineered class B GPCR–Gs–βarr1 megaplex, no direct interaction between the heterotrimeric Gs and βarr1 was observed, and thus, it is not obvious how βarrs may affect G protein activity from this structure (*Nguyen et al., 2019*). On the other hand, biochemical studies of the megaplex and G protein–βarrs interactions demonstrated that βarr can serve as a scaffold for the Gβγ subunits that are released upon activation of the heterotrimeric G protein (*Thomsen et al., 2016*; *Wehbi et al., 2013*; *Yang et al., 2009*). Thus, this Gβγ scaffolding role of βarr may confine $G\alpha_s$ and $G\alpha_{q/11}$ near endosomally located $V_2R$, leading to their re-activation as soon as the inactive GDP-bound Gα with Gβγ subunits reassemble. The results of such activation mechanism would be a net increase in the G protein activation rate.

In cells, confocal microscopy imaging and proximity assays such as BRET have been used to detect megaplex formation at class B GPCRs including the $V_2R$ (*Thomsen et al., 2016*). However, these experimental approaches as well as other techniques that detect protein associations in intact cells cannot discriminate between direct protein–protein interactions and proteins that are in close proximity to each other yet not physically connected. Therefore, the exact protein configuration of detected megaplexes is extremely difficult to confirm in cells, and there is a theoretical possibility that the responses measured and interpreted as megaplexes formation may arise from receptor dimers where each individual receptor binds either G protein, βarr, or both at the same time. To confirm that a single receptor in cells can physically interact with G protein and βarr simultaneously, we previously conducted a series of experiments with class B GPCR–βarr fusions where the two proteins are covalently attached at the receptor carboxy-terminal tail. We showed that despite the GPCR–βarr coupling being fully functional and this fusion 'complex' internalizing constitutively, the receptor maintains its ability to activate G proteins upon agonist stimulation (*Nguyen et al., 2019*; *Thomsen et al., 2016*). Thus, these results definitively showed that single receptor megaplexes can *physically* form in cells.

Surprisingly, our results using Δβarr1/2 cells indicate that the $V_2R$ not only promote endosomal G protein signaling in a βarr/megaplex-dependent manner but also can internalize and activate G proteins from endosomes in a βarr-independent fashion (*Figure 4B C*). Although our data showed that βarr-independent endosomal G protein activation is substantial less effective than the βarr-dependent mechanism for the $V_2R$, it still represents an alternative mode of endosomal GPCR signaling that little is known about. For some receptors such as the apelin receptor βarr association is not required for clathrin-mediated internalization (*Pope et al., 2016*). Alternatively, several GPCRs also internalize independently of any clathrin and βarr associations via caveolae or by fast endophilin-mediated endocytosis (*Moo et al., 2021*). Notably, the glucagon-like peptide-1 receptor (GLP1R) internalizes in a βarr-independent manner and signals via $G\alpha_s$ from endosomes to promote glucose-stimulated insulin secretion in pancreatic β cells, which highlights the physiological relevance of βarr-independent endosomal G protein signaling (*Claing et al., 2000*; *Kuna et al., 2013*). Interestingly, in a very recent study of the vasoactive intestinal peptide receptor 1 (VIPR1) by Blythe and von Zastrow, it was shown that

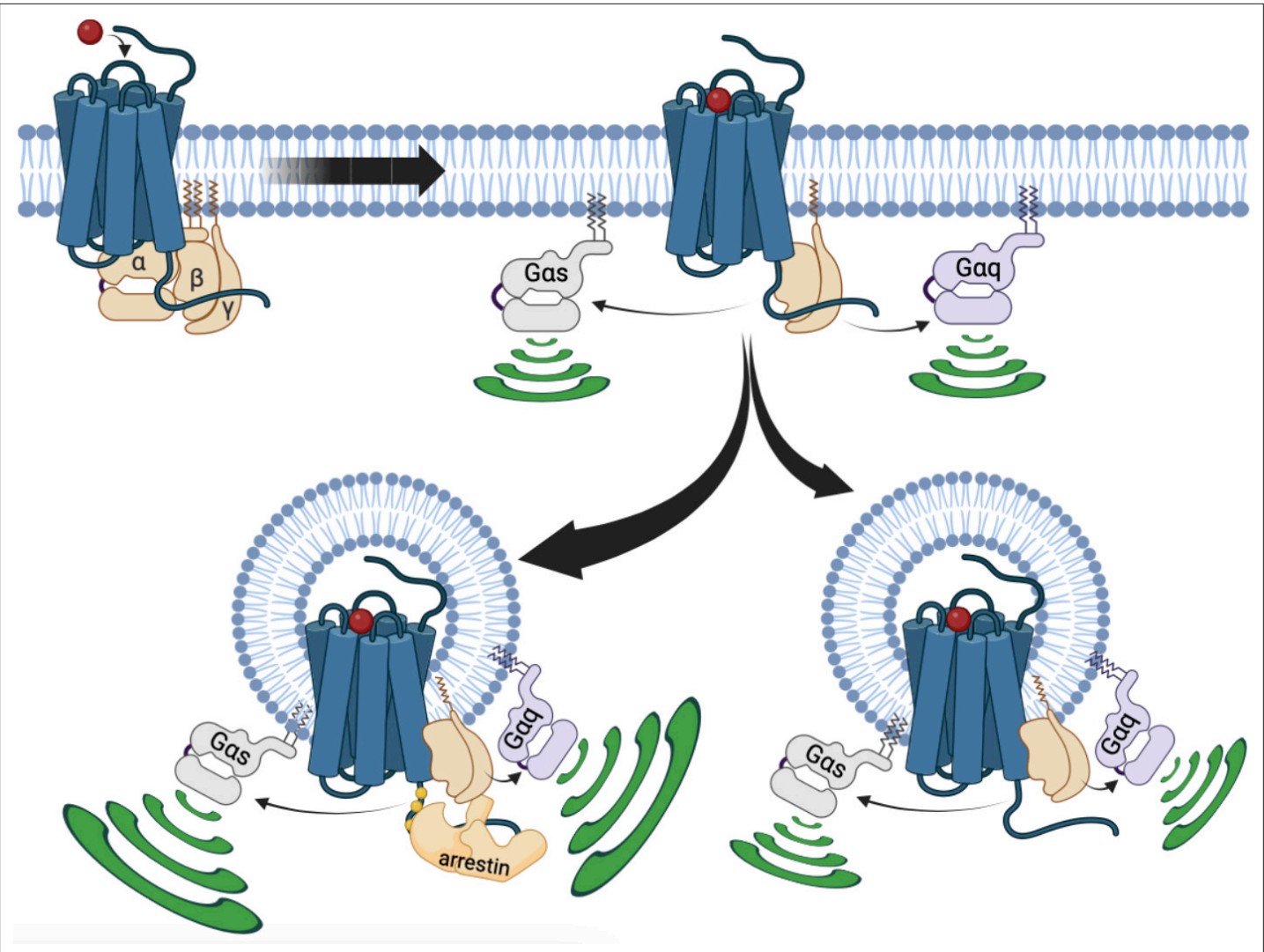

**Figure 5.** Updated model of vasopressin type 2 receptor (V$_2$R) signaling. At the plasma membrane, arginine vasopressin (AVP) binding to V$_2$R results in receptor-mediated Gα$_s$ or Gα$_q$ activation. This initial G protein activation at the plasma membrane is followed by V$_2$R internalization into early endosomes. This internalization occurs primarily in a βarr-dependent manner, leading to the formation of a megaplex with Gα$_s$ or Gα$_{q/11}$ and robust activation of these G proteins from endosomes. Additionally, a minor population of V$_2$R internalize in a βarr-independent fashion, which also leads to minor but significant Gα$_s$ or Gα$_{q/11}$ activation from endosomes.

VIPR1 promotes robust G protein signaling from endosomes and that this occurs in a completely βarr-independent fashion (***Blythe and von Zastrow, 2023***). Surprisingly, the authors observed that agonist stimulation of VIPR1 led to recruitment of βarr1 and receptor internalization into endosomes where VIPR1 and βarr1 colocalized. However, despite this potential interaction between VIPR1 and βarrs, the presence of βarr1/2 had little to no effect on receptor internalization and the ability of VIPR1 to activate G protein from endosomes (***Blythe and von Zastrow, 2023***). As three independent studies using two different receptor systems now have found that endosomal G protein signaling can be achieved independent of βarrs, it is likely that this alternative mode of signaling represents a more general mechanism that is utilized by multiple GPCRs to regulate important physiological functions. Thus, further investigation into the details of βarr-independent receptor internalization and endosomal G protein signaling is much needed.

In summary, in the present study we gain new insights into how internalized V$_2$R stimulates G protein signaling from endosomes, which require us to modify the current model (***Figure 5***). We demonstrated that V$_2$R-mediated endosomal G protein activation is not restricted to the Gα$_s$ isoform but also occurs with the Gα$_{q/11}$ isoforms. A major part of this endosomal G protein activation is βarr-dependent, and

presumably takes place through the formation of $V_2R$–G protein–βarr megaplexes. Interestingly, the presence of βarrs in these megaplexes potentiates the ability of the $V_2R$ to activate G protein within endosomes. Surprisingly, we found that this mechanism is not the only way internalized $V_2R$ stimulates G protein signaling from endosomes since this event can take place in a completely βarr-independent fashion as well. The underlying details of how βarr-independent endosomal G protein activation by the $V_2R$ takes place is not known. However, since similar observations were made of other GPCRs including the GLP1R and VIPR1, the mechanism might represent a general aspect of GPCR biology that control important physiological and pathophysiological processes.

## Materials and methods

### Cell culture and transfection

HEK293 clonal cell line (HEK293SL cells) and referred as HEK293 cells as well as the HEK293 cells devoid of βarr1 and βarr2 referred as Δβarr1/2 cells were a gift from Stephane Laporte (McGill University, Montreal, Quebec, Canada), previously described and authenticated by STR (*Namkung et al., 2016*). These cells were cultured in Dulbecco's modified Eagle's medium high glucose (Life Technologies, Paisley, UK) supplemented with 10% fetal bovine serum and 100 units per ml penicillin–streptomycin (Life Technologies, Paisley, UK), maintained at 37°C and 5% $CO_2$ and passaged every 3–4 days using trypsin–EDTA = Ethylenediaminetetraacetic acid 0.05% (Life Technologies, Paisley, UK) to detach the cells. All cells were monthly checked for mycoplasma contamination and were negative. DNA to be transfected was combined with salmon sperm DNA (Thermo Fisher Scientific, Cambridge, UK) to obtain a total of 1 µg DNA per condition. Linear polyethyleneimine 25 K (PEI; Polysciences Europe GmbH, Germany) was combined with DNA (3 µg PEI per µg of DNA), vortexed and incubated 20 min before adding a cell suspension containing 300,000 cells per ml (1.2 ml of cells per condition). The appropriate volume of cells containing the DNA was seeded and cells were incubated for 48 hr before assay.

### DNA plasmids

All DNA constructs were cloned into pcDNA3.1(+) expression plasmid except if stated otherwise. $V_2R$ and $V_2β_2AR$ were tagged with a HA epitope in amino-terminal of the receptors. HA-$V_2R$ was synthetized by GenScript and HA-$V_2β_2AR$ was generously provided by Dr Robert Lefkowitz (Duke University, USA). C-tRFP-Lck (cloned into PCMV6-AC-RFP expression vector) and TagRFP-T-EEA1 (cloned into pEGFP-C1 vector) were purchased from Addgene (respectively #RC100049 and #42635). Strawberry-βarr2 was a gift from Prof. Marc G. Caron (Duke University, USA). rGFP-CAAX (*Namkung et al., 2016*), rGFP-Rab5 (*Namkung et al., 2016*), $V_2R$-Rluc (*Namkung et al., 2016*), Rluc-βarr1 (*Zimmerman et al., 2012*), Rluc-βarr2 (*Quoyer et al., 2013*), Rluc-C1b (*Wright et al., 2021*), and the BRET-based unimolecular PKC sensor (*Namkung et al., 2018*) were previously described. Rluc-mGs, Rluc-mGsi, Rluc-mGsq, and Rluc-mG12 were synthetized by Twist Bioscience and cloned into pTwistCMV expression vector. The Venus tag in NES-Venus-mGs, NES-Venus-mGsi, NES-Venus-mGsq, and NES-Venus-mG12 previously described (*Wan et al., 2018*) was replaced by Rluc. Halo-tagged mG constructs was kindly provided by Prof. Nevin A. Lambert (Augusta University, USA). LgBiT-mGsq and SmBiT-βarr1 were synthetized by GenScript. mGsq and βarr1 were tagged in amino-terminal with LgBiT and a linker peptide and SmBiT, respectively.

### BRET assays

The cell suspension containing DNA (BRET/EbBRET biosensors and receptors) were seeded in white 96-well plates (Greiner, Stonehouse, UK) at 30,000 cells/well (100 µl per well). Forty-eight hours after transfection, cells were washed with DPBS = Dulbecco's phosphate buffered saline (Life Technologies, Paisley, UK) and assayed in Tyrode's buffer containing 137 mM NaCl, 0.9 mM KCl, 1 mM $MgCl_2$, 11.9 mM $NaHCO_3$, 3.6 mM $NaH_2PO_4$, 25 mM HEPES = 4-(2-hydroxyethyl)-1-piperazineethanesulfonic acid, 5.5 mM glucose, 1 mM $CaCl_2$ (pH 7.4) at 37°C. AVP or vehicle (water) were added and cells incubated at 37°C for the required time. Five minutes before reading, 2.5 µM of the Rluc substrate coelenterazine 400a (NanoLight Technology, Pinetop, AZ, USA) was added. All BRET and EbBRET measurements were performed using a FLUOstar Omega microplate reader (BMG Labtech, Ortenberg, Germany) with an acceptor filter (515 ± 30 nm) and donor filter (410 ± 80 nm). BRET/EbBRET

values were determined by calculating the ratio of the light intensity emitted by the acceptor over the light intensity emitted by the donor. ΔEbBRET is defined as the values of EbBRET in presence of AVP (Merck Life Science, Gillingham, UK) minus the value obtained with vehicle. Dose–response curves were fitted using nonlinear regression using a 4-parameter equation and the basal ΔEbBRET was fixed to zero. Statistical significance of parameters of dose–response curves (AVP-induced maximal efficacy or potency) was established by comparing independent fits with a global fit that shares the selected parameter using extra sum-of-squares $F$-test. The transduction coefficients log($\tau$/Ka) were determined using the operational from Kenakin and Christopoulos as previously described (*van der Westhuizen et al., 2014*).

## NanoBiT assay

The nanoBiT assay to measure proximity between LgBiT-mG proteins and SmBiT-βarr1 has been reported previously (*Hahn et al., 2022*). In short, 2,000,000 cells were seeded per well in 6-well plates. Twenty-four hours later, 125 ng SmBiT-βarr1, 1000 ng $V_2R$ or $V_2\beta_2AR$, and 125 ng LgBiT-mGs or 1000 ng LgBiT-mGsq were transfected into the cells using Lipofectamine 3000 transfection reagent (Thermo Fisher Scientific, Waltham, MA, USA). The next day, transfected cells were detached and 100,000 cells/well were plated into a Poly-D-lysine-coated white 96-well Microplate (Corning, Corning, NY, USA) and incubated overnight at 37°C. The cells were equilibrated in Opti-MEM (Life Technologies, Paisley, UK) at 37°C for 60 min. Coelenterazine-h (NanoLight Technology, Pinetop, AZ, USA) was added at a final concentration of 10 µM before starting the measurement. After establishing a baseline response for 2 min, cells were stimulated with AVP added at a final concentration of 100 nM and the luminescence was measured for additional 20 min. The signal was detected at 550 nm using a PHERAstar *FSX* instrument (BMG LabTech, Cary, NC, USA). ΔRLU is defined as the values of relative luminescence in presence of AVP minus the value obtained with vehicle.

## Confocal microscopy

Cells containing plasmid DNA encoding for fluorescent-tagged localization markers or strawberry-βarr2, receptors, and mGs, mGsq, or mGsi fused to HaloTag were seeded in 8-well glass chambered slides (Thistle Scientific, Uddingston, Glasgow, UK) at 30,000 cells per well. The day of the assay, mG constructs were labeled by adding HaloTag Oregon Green Ligand (Promega, Chilworth, UK) to cells at a final concentration of 1 µM in the culture media and incubated 15 min (37°C, 5% $CO_2$). Cells were washed three times with the media and incubated 30 min (37°C, 5% $CO_2$) for the last wash. The media was aspirated, replaced by Tyrode's buffer and cells were stimulated with AVP or vehicle (water) for the required time at 37°C, 5% $CO_2$. At the end of the incubation, the media was aspirated and cells were fixed by adding 300 µl per well of 4% paraformaldehyde in PBS (Life Technologies, Paisley, UK) and incubated at room temperature for 10 min. The paraformaldehyde solution was aspirated, replaced by DPBS and cells were incubated for 10 min before being replaced by Tyrode's buffer (300 µl per well) and visualized on a SP8 confocal microscope (Leica Biosystems, Nussloch, Germany) at ×63 magnification. Images were quantified using Imaris cell imaging software version 9.9.1 (Oxford Instruments, Abingdon, UK). For each image, a threshold for the red channel was established by selecting the lower intensity from the region of interest (plasma membrane, endosomes, or βarr2). The percentage of colocalization was determined by the percentage of the material from the red channel above threshold colocalized with the material from the green channel. To quantify the percentage of colocalization for confocal microscopy images, the 'surfaces' module was selected isolating cells containing the region of interest for each image. Thresholds for each channel were established by selecting the lower intensity from the region of interest (plasma membrane, endosomes, or βarr2). Data are reported as red volume (red voxels) above the threshold that is colocalized with green volume (green voxels) above the threshold and reported as percentage.

## Enzyme-linked immunosorbent assay

To measure the relative cell surface expression of $V_2R$ and $V_2\beta_2AR$ (both tagged with a HA epitope at their amino-terminal), the same cell suspension containing DNA that was used for EbBRET assays was seeded in white 96-well plates previously coated with Poly-D-lysine (Bio-Techne, Abingdon, UK) at 30,000 cells/well (100 µl per well). Non-transfected cells were used to establish the background of the assay. For the coating, the Poly-D-lysine solution (0.1 mg per ml) was added (50 µl per well)

and the plates incubated at 37 °C for at least 30 min. Following the incubation, the solution was aspirated and wells washed two times with DPBS before adding the cell suspension containing DNA. Forty-eight hours after seeding, cells were washed with DPBS and fixed by adding 50 μl per well of 4% paraformaldehyde in PBS and incubated at room temperature for 10 min. The fixing solution was aspirated and wells washed three times in a washing buffer composed of DPBS containing 0.5% of bovine serum albumin (Merck Life Science, Gillingham, UK). The washing buffer was left in the wells for 10 min following the last wash. After the 10-min incubation, the buffer was removed and 50 μl per well of monoclonal 3F10 anti-HA-Peroxidase (cat# 12013819001, Merck Life Science, Gillingham, UK) 12.5 ng per ml in washing buffer was added and the plate incubated 1 hr at room temperature. The antibody was aspirated and wells washed three times with the washing buffer. The washing buffer was left in the wells for 10 min following the last wash and wells were washed again three times with DBPS only. After aspiration of the DPBS, 100 μl per well of SigmaFast OPD (Merck Life Science, Gillingham, UK) solution prepared as recommended by the manufacturer was added. Wells were incubated in presence of the OPD solution until the wells containing cells expressing receptors become yellow (typically 10 min). The reaction was stopped by addition of 25 μl per well of hydrochloride 3 M in water. 100 μl per well were transferred to a transparent clear 96-well flat bottom plate (Thermo Fisher Scientific, Cambridge, UK) and absorbance at 492 nm was measured using a FLUOstar Omega micro-plate reader. The net absorbance represents the absorbance measured in presence of receptor minus the background (i.e. absorbance measured in absence of receptor).

## Data processing and statistical analyses

In all experiments at least three independent experiments were performed. *n* value is provided in the corresponding figure legend. All experiments were performed in quadruplicates. A p-value ≤0.05 was considered as statistically significant for all analyses. Normally distributed and normalized data to control for unwanted sources of variation are shown as mean ± standard error on mean. All statistical analyses and nonlinear regressions were performed using GraphPad Prism 9.4.1 software.

## Funding information

This work was supported by a Wellcome Trust Seed Award (215229/Z/19/Z) and a New Investigator Award from the UKRI Biotechnology and Biological Sciences Research Council (BB/X002578/1) to BP, a research grant from the LEO Foundation (LF18043) and the NIH (1R35GM147088 and 1R21CA243052) to ARBT, a Doctoral Studentship from the Department for the Economy (DfE) Northern Ireland to CD, a Fellowship from the Armagh Tigers Charitable Trust to AW, and a CITI-GENS Horizon2020 Marie Sklodowska-Curie Doctoral Scholarship to AAG.

## Materials availability statement

The DNA plasmids used in this study can be shared upon request to the corresponding authors, except for the plasmids covered by a MTA from Professors Michel Bouvier and Stéphane Laporte.

## Acknowledgements

We would like to thank Stéphane Laporte for providing the parental and CRISPR/Cas9-engineered β-arrestin1/2-deficient HEK293 cells, Michel Bouvier for providing the rGFP-CAAX and rGFP-Rab5 EbBRET biosensors as well as the BRET-based PKC sensor.

## Additional information

### Funding

| Funder | Grant reference number | Author |
|---|---|---|
| Wellcome Trust | 215229/Z/19/Z | Bianca Plouffe |
| Biotechnology and Biological Sciences Research Council | BB/X002578/1 | Bianca Plouffe |

| Funder | Grant reference number | Author |
|---|---|---|
| LEO Fondet | LF18043 | Alex Rojas Bie Thomsen |
| National Institutes of Health | 1R35GM147088 | Alex Rojas Bie Thomsen |
| National Institutes of Health | 1R21CA243052 | Alex Rojas Bie Thomsen |
| Department for the Economy | | Carole Daly |
| Armagh Tigers Charitable Trust | | Adam Wright |
| CITIGENS Horizon 2020 Marie Sklodowska-Curie | | Akim Abdul Guseinov |

The funders had no role in study design, data collection, and interpretation, or the decision to submit the work for publication. For the purpose of Open Access, the authors have applied a CC BY public copyright license to any Author Accepted Manuscript version arising from this submission.

## Author contributions
Carole Daly, Data curation, Formal analysis, Funding acquisition, Investigation, Writing – original draft; Akim Abdul Guseinov, Adam Wright, Data curation, Formal analysis, Funding acquisition, Investigation; Hyunggu Hahn, Data curation, Formal analysis, Investigation; Irina G Tikhonova, Supervision, Writing – review and editing; Alex Rojas Bie Thomsen, Conceptualization, Data curation, Formal analysis, Supervision, Funding acquisition, Investigation, Visualization, Writing – review and editing; Bianca Plouffe, Conceptualization, Data curation, Formal analysis, Supervision, Funding acquisition, Investigation, Visualization, Project administration, Writing – review and editing

## Author ORCIDs
Bianca Plouffe (iD) http://orcid.org/0000-0002-8321-0796

Joint Public Review: https://doi.org/10.7554/eLife.87754.3.sa1
Author Response https://doi.org/10.7554/eLife.87754.3.sa2

# Additional files

## Supplementary files
• MDAR checklist

## Data availability
All data generated or analyzed during this study are included in the manuscript and supporting files; Source data files have been provided for Figures 1–4.

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
