## [Editor Report · eLife assessment]

This is an **important** study that contributes to our understanding of the role of beta-arrestins in endosomal activation of the vasopressin type 2 receptors. While the use of a minigene as a tool is a weakness, the evidence is overall **convincing** and makes for significant findings whose theoretical and practical implications extend to other GPCRs.

---

## [Referee Report · Joint Public Review]

The authors present a carefully controlled set of experiments that demonstrate an additional complexity for GPCR signaling in that endosomal signaling may be different when beta-arrestin is or isn't associated with a G protein-bound V2 vasopressin receptor. It uses state of the art biosensor-based approaches and beta-arrestin KO lines to assess this. It adds to a growing body of evidence that G proteins and beta-arresting can associate with GPCR complexes simultaneously. They also demonstrate the possibility that Gq might also be activated by the V2 receptor. My sense is one thing they may need to be considered is the possibility of such "megacomplexes" might actually involve receptor dimers or oligomers. They have added significant amounts of new data to address concerns I had about the sole use of mini-genes to assess G protein coupling and broadened discussions about the possible mechanisms underlying their observations. I would still argue that receptor oligomers are a more obvious way for such megacomplexes to be organized around.

---

## [Author Response]

The following is the authors’ response to the original reviews.

**REVIEWER #1:**
The authors present a carefully controlled set of experiments that demonstrate an additional complexity for GPCR signaling in that endosomal signaling make be different when b-arrestin is or isn't associated with a G protein-bound V2R vasopressin receptor. It uses state of the art biosensorbased approaches and b-arrestin KO lines to assess this. It adds to a growing body of evidence that G proteins and b-arrestin can associate with GPCR complexes simultaneously. They also demonstrate the possibility that Gaq might also be activated by the V2R receptor. My sense is one thing they may need to be considered is the possibility of such "megacomplexes" might actually involve receptor dimers or oligomers.1.1 Can the authors please review the data that describes the concept of "GPCR megacomplexes"? I feel this is missing from the introduction. The notion means different things to different people. As you will see from my other comments, you should especially focus on evidence at the level of the single receptor.

We appreciate the reviewer’s comments and have now included a more wholesome description of the GPCR megacomplex, or ‘megaplex’, concept in the introduction (page 2, 1st paragraph).

1.2 The authors use mini-G proteins to conclude that V2R receptors interact with Gaq (in addition to Gas). I would prefer if there were a more direct measure of this. Can the authors show that the receptor interacts with full length Gaq (and not the other G proteins in Figure)? Is there a signaling phenotype associated with Gaq coupling? Is it sensitive to Gaq inhibition?

Excellent point and we are happy to expand further on this. The ability of the V2R to activate Gq/11 has already been demonstrated before (Zhu, X. et al. Mol Pharmacol 46(3):460-9 (1994); Lykke, K. et al. Physiol Rep. 3(8):e12519 (2015); Avet, C. et al. eLife 11: e74101 (2022); Heydenreich, F.M. et al. Mol Pharmacol 102(3):139-49 (2022)). Therefore, we did not attempt to document this activation using more traditional assays. On the other hand, to demonstrate an interaction between V2R and Ga subunit in cells is challenging for several reasons. First, the full-length Ga subunit is already located at the plasma membrane at basal state, and thus, generates high background signals in proximity assays. Second, upon receptor activation, the Ga subunit interaction with V2R is so transient that it is difficult, if not impossible, to catch this transient moment in a proximity assay. Although the miniG proteins are highly engineered, coupling specificity of the different subtypes (Gas, Gai/o, Gaq/11, and Ga12/13) to GPCRs is maintained. In addition, as they are homogenously expressed in the cytosol under basal states rather than at the membrane, they generate low background noise. Upon agonist stimulation, miniG proteins are recruited from the cytosol to the V2R at the plasma membrane, resulting in a robust signal in proximity assays. Thus, miniG proteins are unique in that they can actually detect GPCR–G protein interactions in cellular proximity assays, which is very challenging using full-length Ga subunits.

That being said, we fully understand the reviewer’s concern and greatly value the effort in enhancing robustness of our study. Therefore, we have now monitored downstream signaling events of Gaq/11 in the absence or presence of the selective Gaq/11 inhibitor YM-254890 as a secondary method of documenting Gaq/11 activity. Specifically, we used a newly developed biosensor to measure diacylglycerol (DAG) production, a downstream second messenger of Gaq/11 activation, at both the plasma membrane and endosomes. Using a second biosensor, we detect general protein kinase C (PKC) activation, which is another downstream signaling event of Gaq/11 activation. Together, we demonstrated that AVP-stimulation leads to DAG production at both the plasma membrane and endosomes (Fig. 1C-D) as well as PKC activation (Fig. 1E), which all are sensitive to YM-254890 inhibition (Fig. 1C-D and E). Together these results rigorously suggest that the V2R interacts with and activates Gaq/11.

1.3 I raise a similar concern with Gaq coupling in endosomes.

For similar reasons that miniG proteins are excellent tools for demonstrating V2R interaction with G proteins at the plasma membrane, miniG proteins can also be used to detect V2R interaction with G proteins at endosomes by measuring proximity between miniG and an endosomal marker in response to agonist challenge. However, to ensure that the endosomal recruitment of miniGsq to the V2R demonstrated in our study corresponds to endosomal Gaq/11 activation, we monitored the production of DAG at the early endosomes in a similar way to which we detected DAG production at the plasma membrane. As shown in Fig. 1D, stimulation of V2R with AVP induces recruitment of the DAG-binding biosensor to the early endosomal marker Rab5. Pre-treatment of the cells with the selective Gaq/11 inhibitor YM-254890 abrogated this response, confirming that V2R activation leads to production of DAG at the early endosomes in a Gaq/11-dependent manner (Fig. 1D).

1.4 Can the confocal data be shown for Gai and Ga12?

Yes, we can certainly show this data as negative control. We have now included the confocal data using Halo-mGsi as a negative control for confocal microscopy (Fig. 2). As seen on this figure, mGsi does not colocalize with Lck (plasma membrane), nor with EEA1 (early endosomes) upon stimulation of cells with AVP in line with a receptor that does not couple to Gai/o.

We did not include data using Halo-mG12, as this G protein subtype, similar to Gi/o, does not couple functionally to V2R. Therefore, it is highly unlikely we would obtain different results from the experiments using Halo-mGsi.

1.5 The authors want us to believe that there is simultaneous binding of G proteins and b-arrestin. This is never demonstrated and is at odds with the structural basis of G protein and b-arrestin binding. Have the authors considered that "simultaneous" occupancy might simply reflect binding at distinct GPCR monomers in the context of dimeric or oligomeric receptors? They could I suppose provide data at the level of a single receptor rather than using the bulk BRET approaches used.

We appreciate the comment and opportunity to highlight some of our previous work, which address the megacomplexes at the level of a single receptor. First, we have characterized the megacomplex biochemically and structurally at a low resolution (Thomsen ARB et al. 2016, Cell 166(4):907-19). The results unequivocally demonstrate that a single GPCR interacts simultaneously with heterotrimeric G protein, at the receptor core, and with b-arrestin via the phosphorylated receptor carboxy-terminal. We also documented functionality of the megacomplex as the receptor can interact with and activate the G protein, which were shown by 3 different biochemical approaches (Thomsen ARB et al. 2016, Cell 166(4):907-19). In addition, we solved a high-resolution cryo-EM structure of a megacomplex further highlighting the architecture of this complex (Nguyen AH et al. 2019, Nat Struct Mol Biol 26:1123-31). As both biochemical and structural analyses were done in vitro in which the receptor was embedded in a detergent micelle, we also confirmed that the megacomplex structural architecture fits naturally within the context of a membrane in molecular dynamics simulation experiments (Nguyen AH et al. 2019, Nat Struct Mol Biol 26:1123-31).

In cells, we and others have also showed that GPCRs such as the V2R can bind b-arrestins exclusively via the phosphorylated carboxy-terminal tail as it does in the megacomplex (Kumari P et al. 2016, Nat Commun 7:13416; Cahill III TJ et al. 2017, PNAS 114(10):2562-67; Kumari P et al. 2017, Mol Biol Cell 28(8):1003-10; Chen K et al. 2023), Nature (online doi: https://doi.org/10.1038/s41586-023-06420-x). In addition, we and others have used BRET and confocal microscopy to show that the V2R and other GPCRs recruit G protein and b-arrestin simultaneously and that the three components colocalize in endosomes upon prolonged agonist exposure Thomsen ARB et al. 2016, Cell 166(4):907-19; Chen K et al. 2023, Nature (online doi: https://doi.org/10.1038/s41586-023-06420-x). As the reviewer correctly points out, in these cellular experiments (as well as in single molecule microscopy), the working resolution is not high enough to rule out that the receptors that co-recruit G protein and b-arrestin in endosomes could be dimeric instead of monomeric. Thus, we conducted a series of experiments with GPCR–b-arrestin fusions where the two proteins are covalently attached at the receptor carboxy-terminal tail. We showed that despite the GPCR–b-arrestin coupling being fully functional (in respect to b-arrestin promoting a highaffinity state of the receptor for agonist binding and constitutively internalizing the receptor) the receptor could still activate G proteins (Thomsen ARB et al. 2016, Cell 166(4):907-19; Nguyen AH et al. 2019, Nat Struct Mol Biol 26:1123-31), which demonstrates that the single receptor megaplex can physically form in cells.

We have now included an extra paragraph in the discussion to go over these megaplex-related considerations (5th paragraph in the discussion), and we thank the reviewer for raising this point.

1.6 Please introduce abbreviations when you first use this- this was not done consistently.

Thank you for noticing these errors, which we now have corrected.

**REVIEWER #2:**
This manuscript by Daly et al., probes the emerging paradigm of GPCR signaling from endosomes using the V2R as a model system with an emphasis on Gaq/11 and b-arrestins. The study employs cellular imaging, enzyme complementation assays and energy transfer-based sensors to probe the potential formation of GPCR-G-protein-b-arrestin megaplexes. While the study is certainly very interesting, it appears to be very preliminary at many levels, and clearly requires further development in order to make robust conclusions. The authors should consider expanding on this work further to make the points more convincingly to make the work solid and impactful. The two corresponding authors are among the leaders in the field having demonstrated the existence of megaplexes, and building on the work in a systematic fashion should certainly move the paradigm forward. As the work presented in the current manuscript is already pre-printed, the authors should take this opportunity to present a completer and more comprehensive story to the field.

We are grateful for the time and efforts the reviewer has put into reviewing our work. We are certainly excited to learn that the reviewer finds our work “very interesting”. Regarding the robustness, we have added extra control experiments to increase the completeness of the study. These experiments include:

• Measurements of AVP-stimulated diacylglycerol production, a signaling event downstream of Gaq/11 activation. These measurements were conducted both at plasma membrane (Fig. 1C) and early endosomes (Fig. 1D) using a newly developed DAG-binding biosensor, and demonstrate that the V2R activates Gaq/11 at both of these subcellular locations.

• Monitoring AVP-promoted protein kinase C activation, another downstream signaling effect of Gaq/11 activation (Fig. 1E). The result of this approach shows in another way that V2R activates of Gaq/11.

• Inhibition of signaling events downstream of Gaq/11 activation using the selective of Gaq/11 inhibitor YM254890. YM-254890 inhibits both AVP-stimulated DAG production at plasma membrane and endosomes as well as PKC activation (Fig. 1C-E), which strongly confirms that these signaling outputs are results of Gaq/11 activation.

• We have also included the confocal data using Halo-mGsi as a negative control for confocal microscopy (Fig. 2). As seen in this figure, mGsi does not translocate to the plasma membrane or early endosomes upon stimulation with AVP, which validates that V2R activation does not couple to and activate Gai/o.

Finally, we would like to kindly remind the reviewer that the production of the pre-print manuscript is part of the peer-review process in eLife.

2.1 The use of miniG proteins in these experiments is a major concern as these are highly engineered and may not represent the true features of G proteins. While these have been used as a readout in other publications, their use in demonstrating megaplex formation is sub-optimal, and native, full-length G proteins should be used.

We are a bit unsure as to what the reviewer means by using native full-length G proteins. If the reviewer is suggesting to co-immunoprecipitate V2R with native unlabeled G protein and b-arrestin, it should be considered that the G protein interaction with the receptor is extremely transient and unlikely to survive the pull-down procedure unless stabilized by a nanobody or crosslinking. Although the b-arrestin interaction with the receptor is more stable of nature, co-immunoprecipitation with the receptor requires crosslinking or stabilization with a Fab/nanobody. Therefore, we do not think this approach can be used as a more accurate way of detecting native megaplexes.

If the reviewer is suggesting the use of full-length G proteins in our cell-based proximity assays instead of miniG proteins, we would like to highlight that this approach is somewhat prone to false-positive responses. The major reason behind this is that G proteins are located at regions in membranes close to the receptor whereas b-arrestins are distributed throughout the cytosol. Upon activation of the V2R, barrestins translocate to the receptor at the plasma membrane, which results in enhanced BRET between V2R-coupled G protein subtypes and b-arrestins (see Author response image 1 below of preliminary data). This translocation also results in non-specific BRET signals between b-arrestins and G protein subtypes at the plasma membrane that do not couple to V2R but are located in close proximity to the receptor. As these nonspecific BRET signals do not report on the formation of functional V2R megaplexes (see Author response image 1), we have purposely not used this approach.

**Author response image 1. sa2fig1:** 

To overcome this technical hurdle in detection of functional megaplexes, we have replaced full-length G proteins by miniG proteins as the latter are located in the cytosol at resting states and only translocate to the membrane area if a receptor adopts an active conformation. This replacement is advantageous since activation of megaplex-forming receptors such as the V2R results in simultaneous translocation of miniG proteins and b-arrestins from the cytosol to the receptor at the plasma membrane, which produces a highly specific proximity signal (see Author response image 2 below of preliminary data). When stimulating the V2R, we only observe increases in proximity between b-arrestin1 and miniG proteins that are activated by the V2R (miniGs and miniGsq) but not the miniG proteins that are not activated by this receptor (miniGsi and miniG12) (see Author response image 2). Therefore, usage of miniG proteins offers a more accurate experimental approach to detect functional megaplexes as compared to the usage of full-length G proteins.

2.2 The interpretation of complementation (NanoLuc) or proximity (BRET) as evidence of signaling is not appropriate, especially when overexpression system and engineered constructs are being used.

We thank the reviewer for raising this concern. We have previously demonstrated global Gas activation and Gas signaling in form of cAMP stimulated by internalized V2R (Thomsen ARB et al. 2016, Cell 166(4):907-19). As mentioned previously, in the current updated manuscript we have now included experiments to document downstream signaling events in response to Gaq/11 activation. These experiments include measurement of production of DAG at the plasma membrane (Fig. 1C) and early endosomes (Fig. 1D), as well as phosphorylation/activation of PKC (Fig. 1E). Pre-incubation with the selective Gaq/11 inhibitor YM-254890, abrogated all these downstream signals and confirms that the V2R stimulates Gaq/11 protein signaling at both the plasma membrane and endosomes (Fig. 1C-E).

2.3 After the original work from the same corresponding authors on megaplex formation, the major challenge in the field is to demonstrate the existence and relevance of megaplex formation at endogenous levels of components, and the current study focuses solely on showing the proximity of Gaq and b-arrestins.

We completely agree with the reviewer that it will be important to demonstrate functionality endogenous megaplexes and we are currently working on this in other studies using different receptor systems. However, doing this is not trivial and we will have to overcome major technical barriers that we feel is somewhat out of the scope of the current study. The goal of our V2R study is to demonstrate that V2R megaplexes form with Gaq/11 resulting to Gaq/11 activation at endosomes, and that endosomal G protein activation by the V2R can occur independently of b-arrestin, which we in our humble opinion accomplish.

2.4 The study lacks a coherent approach, and the assays are often shifted back and forth between the two b-arrestin isoforms (1 and 2), for example, confocal vs. complementation etc.

We understand the reviewer’s concern. However, as opposed to the β2-adrenergic receptor that binds βarrestin2 with higher affinity than β-arrestin1, V2R has a strong affinity for both β-arrestin1 and β-arrestin2 (Oakley et al. 2000, JBC 275(22):17201-10). The V2R’s almost identical affinity for β-arrestin1 and βarrestin2 is well illustrated in Fig. 3B. Thus, although different β-arrestin isoforms were used in some experiments, it is very unlikely that the overall results and conclusions from this study will change by adding extra experiments to ensure that both β-arrestin isoforms are used in every experiment.

2.5 In every assay, only the G proteins and b-arrestins are monitored without a direct assessment of the presence of receptor, and absent that data, it is difficult to justify calling these entities megaplexes.

Mini G proteins and b-arrestin come into close proximity upon agonist stimulation of the V2R. Using confocal microscopy, we observed this co-recruitment of miniGs/miniGsq and b-arrestin in response to prolonged V2R stimulation at endosomes specifically (Fig. 3D-F). In absence of GPCR stimulation, both miniG and b-arrestin would be homogenously distributed throughout the cytosol, and thus, the only reason to why both proteins have been recruited to endosomes in response to AVP challenge is that they are recruited to internalized and active V2R. This point was obviously not adequately described in the original manuscript, and thus, we have now clarified this further in the updated manuscript at the 8th sentence of the last paragraph of the "The V2R recruits Gas/Gaq and barrs simultaneously" section.

**REVIEWER #3:**
The manuscript by Daly et al. examines endosomal signaling of the vasopressin type 2 receptors using engineered mini G protein (mG proteins) and a number of novel techniques to address if sustained G protein signaling in the endosomal compartment is enhanced by b-arrestin. Employing these interesting techniques they have how V2R could activates Gas and Gaq in the endosomal compartments and how this modulation could occur in arrestin-dependent and -independent manner. Although the phenomenon of endosomal signaling is complex to address the authors have tried their best to examine these using a number of well controlled set of experiments. Though this is an interesting and well carried out study of endosomal signaling of G proteins, my concerns are:3.1 The study is done in overexpressed HEK 293 cells with these engineered constructs making me wonder if the kinetics would be the same in primary cells?

The reviewer raises an interesting and valid point. It is possible that in the context of primary cells the kinetic would differ slightly and it would definitely be interesting to address this in a subsequent study. However, despite being an interesting aspect of our study, the kinetic itself is not our major take home message, but rather the subcellular localization of the G protein activation and the role of β-arrestin in these events. We have now highlighted this aspect in our updated manuscript (1st paragraph of the discussion) and we thank the reviewer for addressing this.

3.2 The use of the phrase "G protein activation independent of b-arrestins to a minor degree" would make me question its physiological relevance. The authors should discuss the relevance of their findings in physiological or pathological context.

We are glad that the reviewer focuses on this point, and we would like to highlight that other GPCRs including the glucagon-like peptide-1 receptor (GLP1R) internalizes in a β-arrestin-independent manner (Claing A et al. 2000 PNAS 97(3):1119-24), while signaling through Gas from endosomes. In the case of the GLP1R, this endosomal Gas signaling promotes glucose-stimulated insulin secretion in pancreatic βcells (Kuna RS et al. 2013 Am J Physiol Endocrinol Metab 305:E161-70). Consequently, β-arrestinindependent endosomal G protein signaling appears to have some physiological relevance. Similarly, in a very recent pre-print from the von Zastrow group (Blythe EE and von Zastrow M 2023 BioRxiv https://doi.org/10.1101/2022.09.07.506997), it was reported that endogenously-expressed vasoactive intestinal peptide receptor 1 (VIPR1), which regulates gastro-intestinal functions, promotes robust G protein signaling from endosomes in a completely β-arrestin-independent fashion. This again suggest that endogenously expressed GPCRs can internalize and activate G proteins from endosomes independently from β-arrestin to produce physiological responses. We have now discussed about these studies in the 6th paragraph of the discussion.

3.3 The confocal colocalization studies shown in Figure 2 and their conclusion "suggesting a certain level of endosomal Gas/Gaq signaling despite the absence of barr2" seems rather inconclusive.

As opposed to V2R a receptor that retains β-arrestin in endosomes upon internalization, β-arrestin quickly dissociates from V2b2AR after internalization due to the low affinity of the carboxy-terminal of β2AR for βarrestin. In the previous Fig. 2 (now Fig. 3), after 45 minutes of AVP stimulation, no β-arrestin is visible at endosomes in cells expressing V2b2AR as β-arrestin has already dissociated from the receptor and translocated back to the cytosol. However, clear green clusters of mGs and mGsq are still visible at endosomes indicating the presence of active receptor interacting with Gas or Gaq despite the fact that βarrestin is back to the cytosol. We quantified the percentage of the green mGs or mGsq clusters that do not colocalize with β-arrestin and have added this information to the updated version of the manuscript (Fig. 3G). In V2R-expressing cells, almost all active receptors that interact with Gas or Gaq/11 also associate with β-arrestin (Fig. 3G). In contrast, in V2b2AR-expressing cells, approximately 75% of the active receptors do not interact with β-arrestin (Fig. 3G). This suggests that β-arrestin binding to V2R is not an absolute requirement for endosomal Gas and Gaq activation by V2R. This point was obviously not addressed adequately in the original manuscript, and thus, we have now elaborated further on this in the updated version in the last paragraph of the "The V2R recruits Gas/Gaq and βarrs simultaneously" section.

3.4 Though a novel observation it is not clear to me how V2R would internalize after activation without arrestin. Is it some sort of generalized microcytosis occurring in these overexpressed cells? Should discuss.

This is certainly a very interesting observation and something other research laboratories also have seen recently – in particular, in context to endosomal G protein signaling (Blythe EE and von Zastrow M 2023 BioRxiv https://doi.org/10.1101/2022.09.07.506997). The main and best characterized pathway for GPCR internalization is clathrin-dependent where receptors most commonly are associated with β-arrestins. However, for some GPCRs, the β-arrestin association is not required for clathrin-mediated internalization. One example is the apelin receptor that can internalize via clathrin-coated pits, but in β-arrestinindependent manner (Pope GR et al. 2016 Moll Cell Endocrinol. 437:108-19). Alternatively, GPCRs can also internalize independently of any clathrin and β-arrestin associations via caveolae or fast endophilinmediated endocytosis (FEME). We have now expanded our discussion of possible mechanisms for βarrestin-independent receptor internalization in the updated manuscript in the 6th paragraph of the discussion, and we thank the reviewer for the suggestion.

3.5 Is use of mini G protein a good representation? The authors should justify.

Excellent point and something we have comprehensively discussed in our response to reviewer 1 and 2 (points 1.2 and 2.1).